# Highly sensitive switching of solid-state luminescence by controlling intersystem crossing

Weijun Zhao[1,2], Zikai He[2,3], Qian Peng[4], Jacky W.Y. Lam[2,5,6], Huili Ma[7,8], Zijie Qiu[2,5,6], Yuncong Chen[2,5,6], Zheng Zhao[2,5,6], Zhigang Shuai[7], Yongqiang Dong[1] & Ben Zhong Tang [2,5,6]

The development of intelligent materials, in particular those showing the highly sensitive mechanoresponsive luminescence (MRL), is desirable but challenging. Here we report a design strategy for constructing high performance On–Off MRL materials by introducing nitrophenyl groups to molecules with aggregation-induced emission (AIE) characteristic. The on–off methodology employed is based on the control of the intersystem crossing (ISC) process. Experimental and theoretical investigations reveal that the nitrophenyl group effectively opens the nonradiative ISC channel to impart the high sensitivity and contrast On–Off behavior. On the other hand, the twisted AIE luminogen core endows enhanced reversibility and reduces the pressure required for the luminescence switching. Thin films can be readily fabricated from the designed materials to allow versatile applications in optical information recording and haptic sensing. The proposed design strategy thus provides a big step to expand the scope of the unique On–Off MRL family.

[1] Beijing Key Laboratory of Energy Conversion and Storage Materials, College of Chemistry, Beijing Normal University, No. 19, XinJieKouWai Street, Beijing 100875, China. [2] Department of Chemistry, Hong Kong Branch of Chinese National Engineering Research Center for Tissue Restoration and Reconstruction, The Hong Kong University of Science and Technology (HKUST), Clear Water Bay, Kowloon, Hong Kong, China. [3] School of Science, Harbin Institute of Technology, Shenzhen, HIT Campus of University Town, Shenzhen 518055, China. [4] Key Laboratory of Organic Solids, Beijing National Laboratory for Molecular Science, Institute of Chemistry, Chinese Academy of Sciences, Beijing 100190, China. [5] NSFC Center for Luminescence from Molecular Aggregates, SCUT-HKUST Joint Research Institutes, State Key Laboratory of Luminescent Materials and Devices, South China University of Technology (SCUT), Guangzhou 510640, China. [6] HKUST Shenzhen Research Institute, No. 9 Yuexing 1st RD, South Area, Hi-tech Park Nanshan, Shenzhen 518055, China. [7] Key Laboratory of Organic OptoElectronics and Molecular Engineering, Department of Chemistry, Tsinghua University, Beijing 100084, China. [8] Key Laboratory of Flexible Electronics (KLOFE) & Institute of Advanced Materials (IAM), Jiangsu National Synergistic Innovation Center for Advanced Materials (SICAM), Nanjing Tech University (NanjingTech), 30 South Puzhu Road, Nanjing 211800, China. These authors contributed equally: Weijun Zhao, Zikai He. Correspondence and requests for materials should be addressed to Z.H. (email: hezikai@hit.edu.cn) or to Q.P. (email: qpeng@iccas.ac.cn) or to Y.D. (email: dongyq@bnu.edu.cn) or to B.Z.T. (email: tangbenz@ust.hk)

Development of intelligent materials[1,2] have drawn continuous attentions for their potential applications in sensors[3–5], displays[6], and memories[7,8]. As one promising candidate, stimulus-responsive materials are highly desirable[9–13]. Of particular interest are those mechanoresponsive luminescent (MRL) materials[14–16], as force is a facile and easily handled external stimulus. From universe to earth, from machine to human body, from cell to organelle, force exists and plays vital roles[17–19]. Moreover, luminescence is a sensitive and visible responsive signal[20], making MRL materials attractive in responding the mechanical properties of local environment and human body[21]. MRL materials provide valuable insights into the artificial intelligent systems and human health status, offering potential applications in data recording and storage, security and counterfeiting, optoelectronic devices and haptic sensors, etc.[22]. As a result, much effort has been placed to develop the MRL family[23–26]. However, the performance of the reported system is still far-reaching from the real applications in terms of sensitivity and reversibility. Examples exhibiting ultra-sensitivity, high contrast, fast and response to low force are even rare[22].

High contrast MRL materials normally show obvious two-color switching or dramatic luminescence intensity change, such as luminescence turn-on and turn-off (On–Off)[27–30]. However, few examples of On–Off MRL materials with high contrast are available in the literature and their overall performance is unsatisfactory. Few limited representatives of On–Off MRL materials utilize the processes of photodimerizaiton[31], photoinduced electron transfer[32,33], intramolecular charge transfer[34,35], and strong π–π interactions modulation[36,37] to control the solid-state luminescence. These traditional strategies employ the strong intermolecular interactions to control the photophysical process of excited state in the solid phase and require strong force to induce the obvious morphology or conformation switching. The reversibility and reproducibility are thus generally low. To realize sensitively on–off luminescence switching, the related photophysical process requires to alter dramatically in the presence of small mechanical stimulus, which is really difficult for traditional MRL materials[23,38,39]. Hence, new design strategies for controlling the related photophysical process in the solid-state of highly sensitive MRL materials are needed but challenging.

Intersystem crossing (ISC) is a nonradiative relaxation process from excited singlet state to highly sensitive triplet state, which weakens or quenches the fluorescence[40–42]. If we can effectively control the ISC process and the triplet state is nonemissive, such an approach can be utilized as a novel strategy to obtain On–Off MRL materials (Fig. 1a). According to the perturbation theory, the rate of ISC, $k_{ISC}$, is influenced by the spin–orbit coupling constant ($\xi_{ST}$) and the energy gap ($\Delta E_{ST}$) between the involved singlet and triplet states, and is expressed by Eq. (1):

$$k_{ISC} \propto \frac{\xi_{ST}^2}{e^{\Delta E_{ST}^2}} \qquad (1)$$

Nitrophenyl group with abundant lone pair electrons can boost the efficient ISC pathway with the aid of a great $\xi_{ST}$ and a

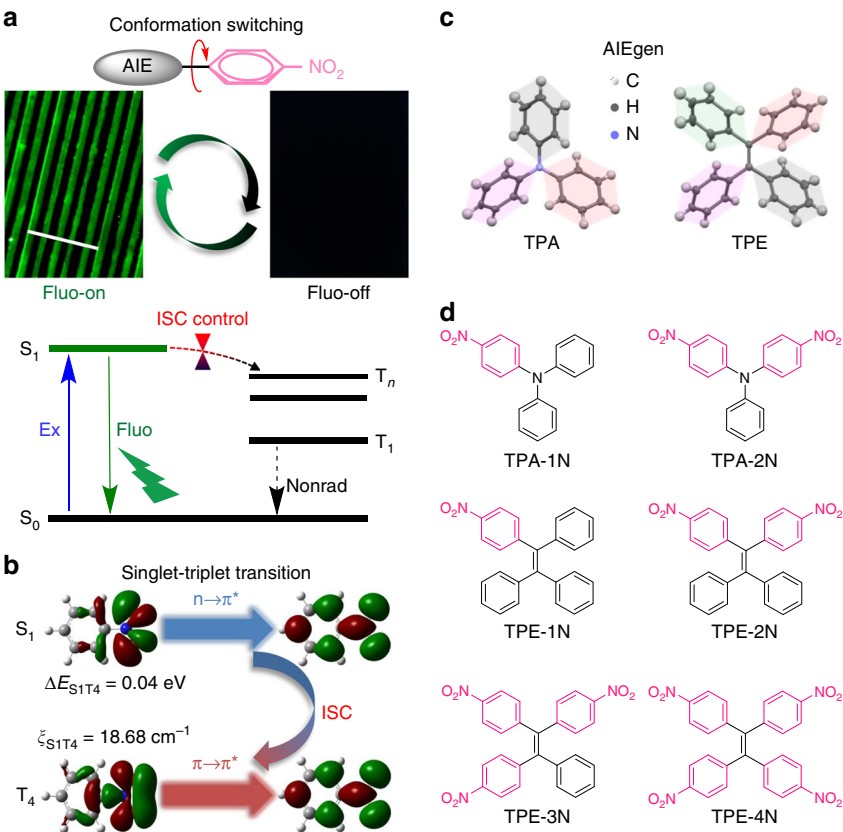

**Fig. 1** Strategies for MRL materials by controlling intersystem crossing. **a** Proposed mechanism of highly sensitive On–Off MRL materials by controlling intersystem crossing of nitrophenylated AIEgens. Scale = 100 μm. **b** Nature transition orbitals of the lowest singlet excited state (S₁) and triplet excited state (T₄) of nitrobenzene. Since El-Sayed's rule states that the multiplicity change becomes highly efficient when the spin–orbit coupling mixes two states differing in both spin and electronic configuration, T₄ is the closet and a good "receiver state" in the intersystem crossing process which determines the ultrafast S₁ depletion through an favored ¹(n,π*) to ³(π,π*) channel. **c** Twisted crystal structures of typical AIEgens: triphenylamine (TPA) and tetraphenyethylene (TPE). **d** Molecular structures of nitro-TPAs and nitro-TPEs studied here

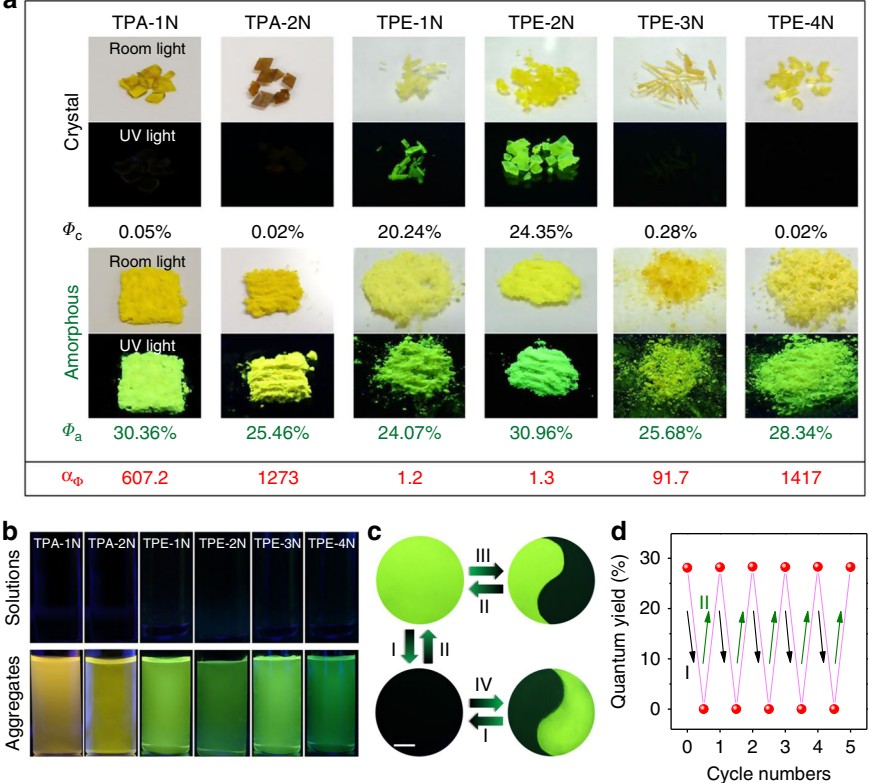

**Fig. 2** Photophysical properties of designed MRL molecules. **a** Photos of luminogens in crystalline and amorphous states taken under room and UV light irradiation with their quantum yields ($\Phi$). Luminescence contrast ratio: $\alpha_\Phi = \Phi_a / \Phi_c$. **b** Fluorescent photos of luminogens in acetonitrile and acetonitrile/water mixtures taken under UV light irradiation, luminogens concentration: $1 \times 10^{-5}$ M. **c** Switching the luminescence of TPE-4N film deposited on a weighing paper through different processes: I, heating at 150 °C or fuming with acetone for 10 s; II, grinding with a glass rod; III, heating with a paper mask for 30 s; IV, grinding with a paper mask. All fluorescent photos were taken under UV irradiation at 365 nm. Scale = 2 cm. **d** Plot of luminescence quantum yields versus repeated grinding and heating cycles

negligible $\Delta E_{ST}$, where the spin–orbit interaction mixes two states differing in both spin and electronic configurations (Fig. 1b, Supplementary Fig. 18)[43–46]. It is well known that $\xi_{ST}$ and $\Delta E_{ST}$ are closely related to molecular conformations and electronic configurations, and thus are highly sensitive to the surrounding environments such as the solid-state morphology. Therefore, it is promising to design new On–Off MRL materials by controlling the ISC process of nitrophenyl-substituted luminophores.

On the other hand, the morphology of organic molecules with twisted conformations can be easily modulated by mechanical stimulus[47,48]. Aggregation-induced emission (AIE) refers to a class of luminogens (AIEgen) which are nonemissive in solutions but become highly emissive when cluster into aggregates[49]. AIEgens often have twisted molecular conformation, such as twisted intramolecular charge transfer conformation[50], *cis–trans* isomerized configuration[51], herringbone conformation[52], etc. Among them, triphenylamine (TPA) and tetraphenyethylene (TPE) are widely used as AIE cores with propeller-like conformation (Fig. 1c). Therefore, AIEgens enjoy intrinsic advantages as excellent sensitive MRL candidates.

In this article, we combine nitrophenyl groups with AIEgens to create sensitive On–Off MRL materials. Six nitro-substituted AIEgens were synthesized and their properties were systemically investigated. The manipulation of ISC is unprecedentedly employed as an on–off methodology and a rational design strategy is introduced to provide a series of high performance On–Off MRL candidates with versatile applications. These results provide a big step in expanding the scope of MRL family.

## Results

**Photophysical property**. To validate our proposal, six luminogens (Fig. 1d) were designed and synthesized through controlling nitration of TPA and TPE. All the molecules were purified by twice recrystallization in a yield of higher than 80% (Supplementary Fig. 1)[53]. As expected, all the six luminogens inherit the AIE property of TPA and TPE core[54]. They are nonemissive in acetonitrile solutions while their aggregates emit bright green or yellow fluorescence (Fig. 2b), which fulfills the basic prerequisite as MRL materials[23].

Morphology-dependent photophysical properties are then carefully investigated (Fig. 2a). In the crystalline state, TPA-1N, TPA-2N, TPE-3N, and TPE-4N are nearly nonemissive with ultralow quantum yields, which indicates that nitrophenyl groups successfully open the nonradiative ISC channel to quench the emission. The ISC becomes the dominant decay pathway as nitrophenyl groups are efficient triplet state promoters[55] (Fig. 1b). In the amorphous state, all the six luminogens recover or enhance the luminescence. They exhibit bright green or yellow fluorescence as well as in thin films and in aggregates, suggesting the ISC is blocked due to morphology change (Supplementary Fig. 2 and Supplementary Table 1-2). The difference is also proved by detecting the morphology-dependent singlet oxygen generated by energy transfer from triplet states (Supplementary Fig. 3). TPA-1N, TPA-2N, TPE-3N, and TPE-4N exhibit the proposed ISC modulated On–Off MRL properties with a high luminescence contrast ratio of up to $10^3$ (Supplementary Figs. 4-6). For comparison, TPE-1N and TPE-2N crystals show intense emission

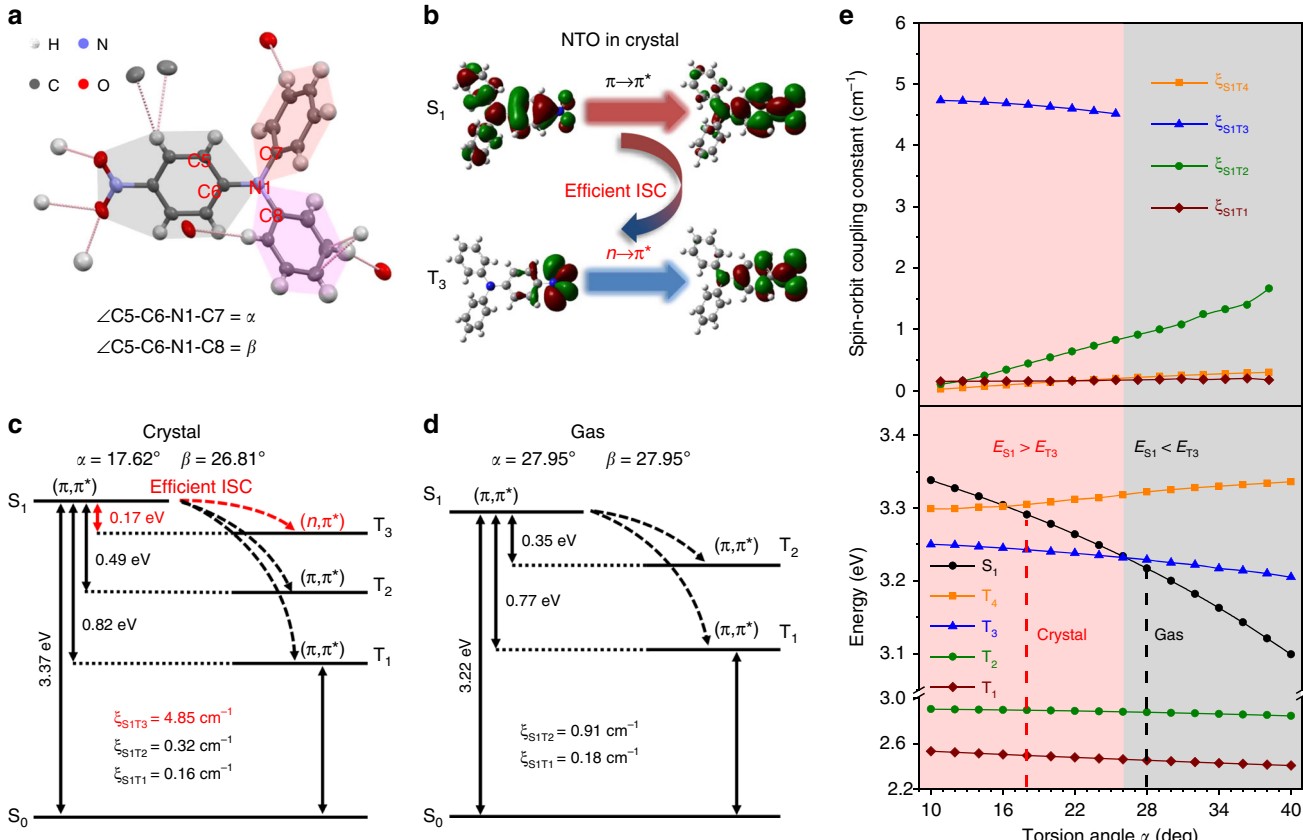

**Fig. 3** Theoretical calculations on the emission mechanism of TPA-1N. **a** Crystal structure and intermolecular interactions between TPA-1N and its adjacent molecules. The torsion angle between nitrophenyl group and other two phenyl groups is assigned to $\alpha$ and $\beta$, respectively. **b** Nature transition orbitals of the lowest singlet excited state ($S_1$) and triplet excited state ($T_3$) of TPA-1N in crystal. **c, d** Calculated energy gaps, electronic configuration character and related spin–orbit coupling constants in crystalline and free gas states. **e** Potential energy curves of $S_1$ and $T_n$ and spin–orbit coupling constants ($\xi$) of related transitions from lowest $S_0$ to $T_n$ as a function of the torsion angle $\alpha$

with high quantum yield of 20.24% and 24.35%, respectively, possibly because the number of nitrophenyl groups is still not enough to overwhelm the AIE effect. Such a factor is critical in constructing On–Off MRL materials, which will be discussed later.

TPE-4N, for example exhibits the On–Off MRL property with the highest contrast. We placed TPE-4N crystals on a piece of weighing paper and ground them into powders, through which a yellow emissive paper was obtained under the UV irradiation. The emission is firstly turned off when annealing at 150 °C or fuming with acetone for 10 s (Process I). The yellow emission can be easily recovered by grinding (Process II). Contrast patterns are obtained through process I and II using a designed mask (Fig. 2c). Monitored by the quantum yields, the processes can be repeated for several cycles without obvious fatigue, suggesting the good reversibility (Fig. 2d and Supplementary Fig. 7).

Organic crystals with twisted molecular structures can be easily amorphized by applying mechanical stimulus. In addition to photophysical study, we performed powder X-ray diffraction (PXRD) and differential scanning calorimetry (DSC) analyses to investigate the morphology change. The PXRD patterns of ground powders show no visible diffraction peaks, indicating their amorphous characteristic caused by mechanical grinding (Supplementary Figs. 8-9). In contrast, the PXRD patterns of thermal annealed or solvent-fumed samples exhibit sharp diffraction peaks which coincide with single crystals, indicating that the amorphous powder transfers to crystalline state. Such transformation is verified by PXRD and DSC in Supplementary Fig. 10. The above results show that the solid-state luminescence

of TPE-4N is strongly related to the morphology, which is easily switched by mechanical stimulus and thermo-annealing or solvent-fuming. Meanwhile, TPA-1N, TPA-2N, and TPE-3N show similar On–Off MRL properties as TPE-4N (Supplementary Figs. 10-12).

Single-crystal XRD analysis shows the presence of abundant weak intermolecular interactions between the nitro and phenyl groups in all the crystals (Supplementary Figs. 13-16). For example, one TPA-1N molecule is surrounded by seven neighboring molecules through multiple weak C–H⋯C and C–H⋯O interactions (Fig. 3a). Together with the twisted structure inheriting from TPA and TPE, those luminogens have quite loose crystal packing. Solvent molecules are encapsulated in the crystal cavities of TPE-3N and TPE-4N (Supplementary Figs. 12, 15, Table 4-6). After removing the solvent molecules, the crystals pack in a looser manner but still are nonemissive, which excludes the possibility of solvent-caused quenching effect (Supplementary Figs. 8-9). For those loose packed crystals, the mechanical force could easily break the weak interactions to release the molecular conformations to free states.

As proposed, AIEgens substituted with suitable nitro groups can realize unique On–Off MRL properties, verifying the validate of our design strategy. However, both crystals and amorphous powders are solid phase, why they exhibit distinctly different luminescence behaviors?

**Theoretical calculation.** To gain deeper insight into the mechanism of On–Off MRL property, we performed

investigations on single molecule in both gas and crystalline states by first-principle density functional theory (DFT) and time-dependent DFT (TD-DFT). The aggregation effect was considered by using the hybrid quantum mechanics and molecular mechanics approach. The computation models are built by digging a $5 \times 5 \times 5$-unit supercell from the single-crystal structures. The calculated energy gaps ($\Delta E_{ST}$), nature transition orbitals, spin–orbit coupling constants ($\xi_{ST}$) between $S_1$ and involved triplet states are summarized in Fig. 3 and Supplementary Figs. 18-25. As well known, luminescence change is the outcome of variation of intermolecular interaction and molecular conformation during morphology transformation.

The effect of intermolecular interaction is first considered. The energy levels and $\xi_{ST}$ are carefully calculated under the same molecular conformation in the absence or presence of intermolecular interactions. The almost same calculation results suggest that the intermolecular interactions are not the main or direct factor for the MRL On–Off properties (Supplementary Figs. 20-25).

The effect of molecular conformation is then investigated. TPA-1N is chosen as the calculated model. As shown in Fig. 3c, d, a big difference in molecular conformation in crystalline and gas state is found, that molecule in crystal has much smaller torsion angles ($\alpha = 17.62°$, $\beta = 26.81°$) than that in free gas state ($\alpha = \beta = 27.95°$). The variation of molecular conformation is accompanied with a decreasing energy gap of $S_1$–$S_0$ from 3.37 eV in crystal state to 3.22 eV in gas state, which agrees with the notable redshift in absorption (Supplementary Fig. 17). Thus, the amorphous state can be considered as a state close to gas state with similar molecular conformation. In detail, TPA-1N molecule in crystal state possesses a typical $^1(\pi,\pi^\star)$ $S_1$ state and a low-lying transition-allowed $^3(n,\pi^\star)$ $T_3$ state with a large $\xi_{S1T3} = 4.85$ cm$^{-1}$ and a small $\Delta E_{S1T3} = 0.17$ eV, which enables the efficient ISC and thus causes fluorescence quenching. For molecule in gas state, the low-lying triplet state changes to a transition-forbidden $^3(\pi,\pi^\star)$ $T_2$ state, accompanied by a reduced $\xi_{S1T2} = 0.91$ cm$^{-1}$ and a larger $\Delta E_{S1T2} = 0.35$ eV, which blocks the efficient ISC process and turns on the fluorescence in amorphous state. Similar explanations are applicable for TPA-2N, TPE-3N, and TPE-4N, where transformation from crystal to amorphous state enlarges $\Delta E_{ST}$ and reduces $\xi_{ST}$, simultaneously (Supplementary Figs. 20-25).

To further simulate how molecular conformation influences the ISC process, a rigid potential energy surface scan is performed by varying the torsion angle $\alpha$ from 10 to 40° in the ground state. The vertical excitation energies of low-lying excited states $S_1$ and $T_{1–4}$ and the involved $\xi_{ST}$ are also calculated. As shown in Fig. 3b, e, when $\alpha$ is smaller than 26°, $T_3$ with a $^3(n,\pi^\star)$ configuration is the closest low-lying triplet state to $S_1$ with a high $\xi_{ST}$ (>4.5 cm$^{-1}$). The different orbital character and the close proximity of the $S_1$ state to the $T_3$ state favors the occurrence of efficient ISC in the crystalline state ($\alpha = 17.62°$). As $\alpha$ is larger than 26°, the energy of $S_1$ decreases significantly which results in $T_3$ higher than $S_1$ in energy. $T_2$ now becomes the closest low-lying triplet state to $S_1$ instead. The ISC then becomes inefficient as $S_1$ and $T_2$ have the same orbital character and a large energy gap exists between them. This situation is similar in the amorphous state.

Last, the effect of substituent number is studied. From TPA-1N to TPA-2N, from TPE-1N to TPE-4N, more nitrophenyl groups gradually increase the number of involved triplet states with minimizing energy gaps and enhancing $\xi_{ST}$ (Supplementary Figs. 20-25). Therefore, the ISC process is boosted by multiple nitrophenyl group substitution to lead TPE-4N the highest On–Off contrast.

**Thin-film On–Off property**. To demonstrate the practical applicability and to measure the sensitivity, TPE-4N in thin film was prepared by simple spin-coating of its chloroform solution on quartz plates. As shown in Fig. 4a and Supplementary Figs. 25-26, the freshly coated film is quite transparent and smooth. It emits bright green light peaked at 520 nm (Fig. 4b). The emission quenches quickly and the PL intensity remarkably decreases by a factor of $10^3$ when the film is annealed at 150 °C by a handed heat gun for 3 s. The PXRD patterns demonstrate that the freshly coated film is amorphous in nature while the annealed one is crystallized (Supplementary Fig. 28). Interestingly, well-defined and bright green emissive words appear when writing on the annealed film by using a fine glass tube. After thermal treatment, the emissive words erase completely. The writing and erasing processes can be repeated for many times, suggesting the excellent reversibility of the fluorescence switching process.

In a further experiment, we demonstrate the On–Off sensitivity of thin-film fluorescence switching. The polarizing optical microscopic photo of the annealed film taken under room light show the presence of uniform spherulites (Fig. 4c). When a weak force is applied using a glass tube, scratch was not observed under room light but was clearly visualized under UV light irradiation with more than 100-fold contrast (Fig. 4d). Again, upon thermal treatment, the bright fluorescent scratch disappears. Because of such ultra-sensitivity, we are now conducting investigations on fluorescence mapping and visualization of stress distribution and fatigue crack propagation for industry use.

**Optical information storage and haptic sensor**. The designed On–Off MRL materials have the advantages of high contrast and sensitivity, good reversibility, and fast response, which encourage us to further explore their practical applications in optical information storage and haptic sensor.

Figure 5a describes the rewriteable optical information storage system containing the procedures of film preparation, micro-embossing, and recovery. A nonemissive film was facilely prepared by thermal annealing of the spin-coated film. After applying a finger pressure using a designed mold to the annealed film, a high-resolution micro-embossing fluorescent patterns can be easily replicated with a width of 10 μm and a spacing of 10 μm. To the best of our knowledge, this is the first MRL patterns with micrometer resolution. The micro-embossing patterns are erased completely upon thermal treatment, and the film is rewriteable again.

Next, we utilized TPE-4N to construct haptic sensor. The TPE-4N films are deposited on aluminum, ceramic, and wooden substrates by simple brush coating process (Methods), which show bright fluorescence (Fig. 5b). The fluorescence signal of the freshly prepared films is turned off completely by thermal annealing. Upon pressing with thumb fingers, the fluorescence of the touched area was turned on immediately to generate well-defined fingerprint patterns observable by naked eyes (Supplementary Fig. 29). In detail, the pressure is gradually enhanced from 0 to 0.98 MPa (Fig. 5c and Supplementary Fig. 30). To allow visualize monitoring, the fluorescent signal is digitized using image grayscale processing. As shown in Fig. 5d, when the applied finger pressure reaches 0.15 MPa, the grayscale intensity increases by 3.5-fold. At 0.25 MPa, a 7.7-fold strong signal was detected. The intensity reaches its maximum (15-fold) at 0.55 MPa. After thermal treatment, the film recovers to its nonemissive state and is reusable (Supplementary Fig. 31). The results suggest a promising fast responsive and reversible haptic sensor.

## Discussion
Nitrophenyl group is a well-known electron and energy acceptor and can serve as an excited state or fluorescence quencher[30,56,57]. However, the use of nitro-substituted aromatics in constructing

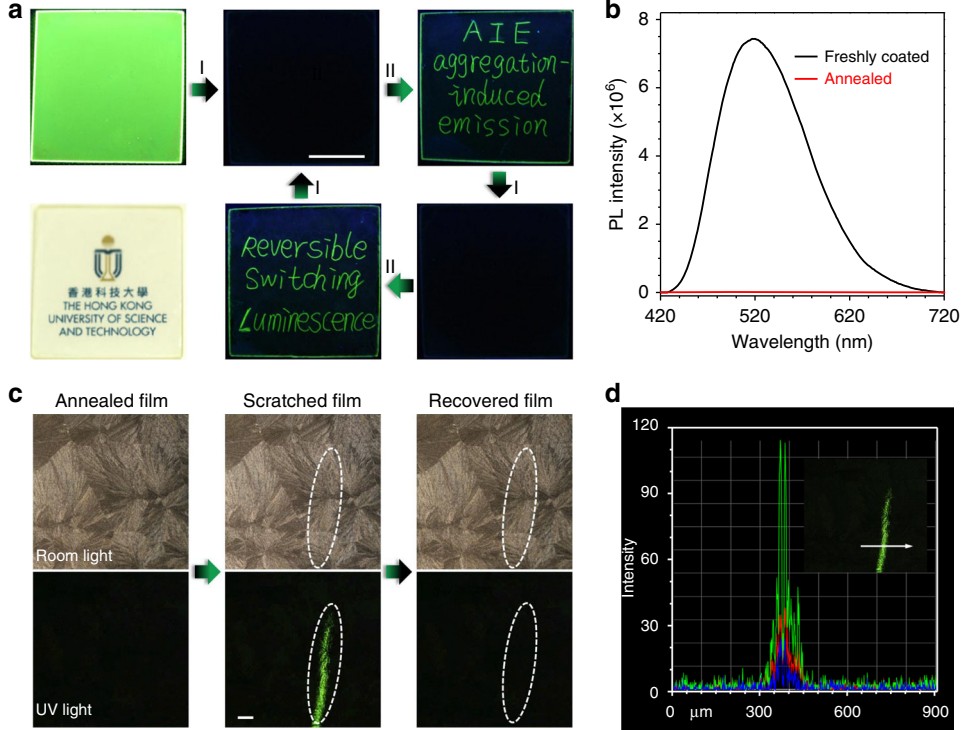

**Fig. 4** High performance On–Off MRL properties of TPE-4N in thin-film state. **a** Luminescent and room-light transparent photos of TPE-4N thin film spin-coated on quartz plate and records of the writing and erasing processes, Process I: heated by a handed heat gun at 150 °C for 3 s; Process II: writing with a fine glass tube. Scale = 1 cm. **b** PL spectra of coated and annealed thin film. **c** Polarizing optical microscopic images of annealed, slightly scratched and recovered film under room light and UV light. Scale = 100 μm. **d** PL intensity contrast spectrum for the slightly ground film in **c**. All the luminescent photos were taken under UV irradiation at 365 nm

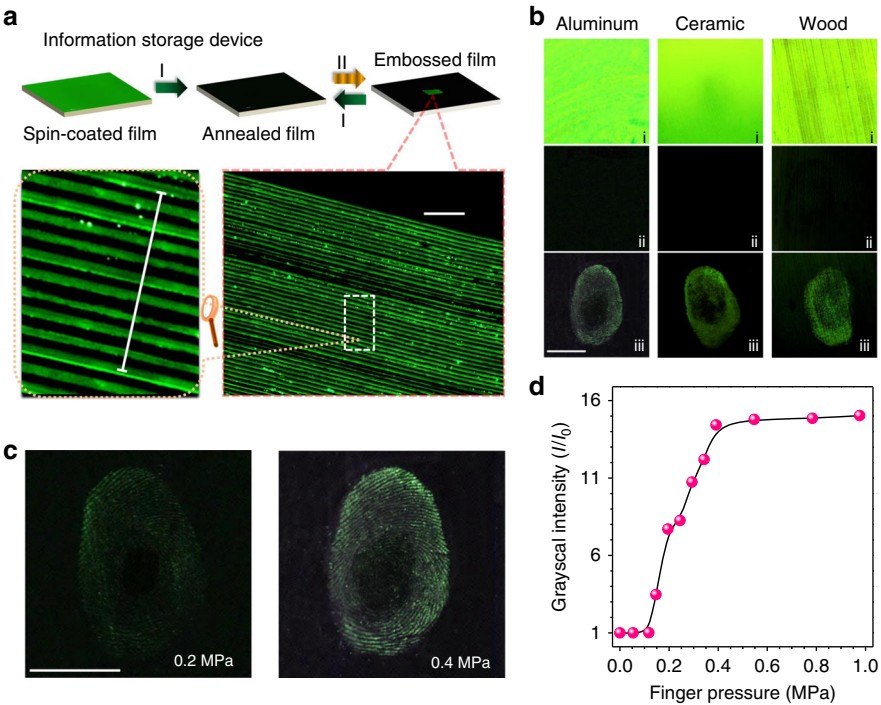

**Fig. 5** Optical information storage and haptic sensor of TPE-4N thin film. **a** Procedures of the micro-embossing and recovery on the thin film of TPE-4N prepared by spin coating on quartz plate and luminescent photos of micro-embossed patterns. Process I: heated by a handed heat gun at 150 °C for 3 s; Process II: embossed with a mold with a width of 10 μm and a spacing of 10 μm. Scale = 200 μm. **b** Haptic photos of fingerprint on aluminum, ceramic, and wooden substrates coated with TPE-4N, i, freshly brush coated film; ii, annealed film; iii, pressed with finger. **c**, **d** Haptic sensor: luminescent photos **c** and grayscale intensity change **d** with the increasing finger pressure. Scale = 1 cm. All luminescent photos were taken under UV irradiation at 365 nm

luminescent materials is seldom explored because it promotes ultrafast nonradiative relaxation ISC process to give nonemissive character of triplet state. Apparently, nitro substitution effect and AIE effect are conflict in terms of solid-state luminescence. Here, we unprecedentedly integrate them in constructing highly sensitive MRL materials. In the crystalline state, the nitrophenyl group effectively opens the nonradiative ISC channel to the highly sensitive and nonemissive triplet states to overwhelm the AIE effect to give fluorescence "Off" state. Mechanical stimulus manipulates the solid-state morphology and changes the molecular conformations. The AIE effect then dominates the photophysical process to recover the solid-state emission to give fluorescence "On" state.

Except ISC, nitrophenyl group imparts abundant weak intermolecular interactions to benefit easy crystallization and loose crystal packing. Coupled with the twisted and propeller-like molecular structure, the morphology transformation of the resulting molecules becomes facile and reversible. The pressure required for luminescence switching is then dramatically reduced to result in excellent sensitivity, fast response, and high contrast On–Off performance. The nitrophenyl group is also helpful for thin-film coating or printing on varied substrate to make the materials valuable for optical information recording and haptic sensing with facile preparation and low cost.

Controlling ISC is a general strategy to achieve novel On–Off MRL materials. Following this strategy, we choose the benzophenonyl group as another triplet promoter and two TPE derivatives, namely TPE-BP and 2TPE-BP, are synthesized as proof-of-concept examples (Supplementary Figs. 32-33). As expected, they are nonemssive in both solution and crystalline state but their amorphous powders emit bright green fluorescence. The benzophenonyl groups are considered to open the efficient nonradiative ISC channels to quench the light emission as they show even stronger triplet-state promotion effect than nitrophenyl group. One benzophenone moiety can turn off the emission two TPE groups in crystalline state. Efforts to expand On–Off MRL family by using this rational design strategy are still in progress.

In summary, we put forth a new design strategy for high performance On–Off MRL materials by controlling the ISC process. Nitro groups are incorporated into twisted AIEgens to facilitate the efficient ISC process. Theoretical calculations indicate that the torsion angle and the number of nitrophenyl group are key factors for tuning the ISC process. Mechanical stimulus modulates the molecular conformations to give varied solid-state morphology with luminescence On–Off switching. Experiments demonstrate the high contrast and sensitive On–Off MRL performance as well as reversibility and fast responding behaviors. Such properties enable them to find versatile applications such as the optical information storage and haptic sensor. Thus, the design strategy introduced in the present study is expected to provide a big step in expanding the scope of On–Off MRL family.

## Methods

**Crystal growth**. All the crystals were cultivated through slow solvent evaporation at room temperature. Single crystals of TPA-1N, TPA-2N, TPE-2N, TPE-3N, and TPE-4N can been obtained easily, while TPE-1N crystal is not suitable for single-crystal analysis. Single crystals of TPA-1N, TPA-2N, and TPE-2N were cultivated from methylene chloride and hexane mixtures. Single crystal of TPE-3N was cultivated from chloroform and hexane mixtures. Single crystal of TPE-4N was cultivated from acetone.

**Solid samples preparation**. The amorphous solids of AIEgens were prepared by heating the crystalline AIEgens to melts with a heating gun and quenching the melts with liquid nitrogen. The ground powers were prepared with a mortar and pestle for 3 min.

**Film preparation of TPE-4N**. Spin-coating films of TPE-4N were prepared by simple spin-coating on quartz substrates ($2 \, cm \times 2 \, cm$) from a chloroform solution (5 wt%). Spin-coating speed: slow spin speed (0.3 kR/min, 6 s) and fast spin speed (1 kR/min, 30 s). Brush coating samples were prepared by simply brushing on different substrates using a chloroform solution (10 wt%).

**Stamping mold**. A stamping mold was prepared on a circular silicon substrate (2 inches in diameter): striping pattern with 0.5 mm in thickness, 10 μm in width, and 10 μm in fringe spacing. The etching depth is 20 μm.

**Data availability**. The authors declare that the all data supporting the findings of this study are available within this article and Supplementary Information files, and also are available from the authors upon reasonable request. CCDC 1550422, 1550423, 1550424, and 1850508 contain the crystallographic data for this paper. These data can be obtained free of charge from The Cambridge Crystallographic Data Centre via www.ccdc.cam.ac.uk/getstructures.

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

## Acknowledgements

The authors acknowledge the financial support by the National Science Foundation of China (51773020, 51703042, 21473214, 21503118, 21290191, 21788102, and 21490574), the Science and Technology Plan of Shenzhen (JCYJ20170811155015918, JCYJ20160509170535223), the Research Grants Council of Hong Kong (16308016, 16305015, C6009-17G, C2014-15G, and A-HKUST605/16), and the Innovation and Technology Commission (ITC-CNERC14SC01 and ITS/254/17) and International Science & Technology Cooperation Program of Guangzhou (201704030069).

## Author contributions

W.Z. synthesized all materials. Z.H. and W.Z. grew the crystals, performed all photophysical measurements and analyses, and prepared the manuscript. Thus, Z.H. and W.Z. contributed equally to this work. Q.P. and J.W.Y.L. revised and edited the manuscript. H.M., Q.P., and Z.S. performed the theoretical calculations. Z.Q., Y.C., and Z.Z. assisted the photophysical property measurement. Y.D., Z.H., Q.P., and B.Z.T. designed and supervised the research and wrote the paper. All authors discussed the results and commented on the manuscript.

## Additional information

**Competing interests:** The authors declare no competing interests.

