## [Peer Review File · Nature Communications]

Reviewers' Comments:

Reviewer #1 (Remarks to the Author):

In the work, the authors put forth a new strategy to design high performance ON/OFF mechanoresponsive luminescent materials by introducing nitrophenyl groups to AIE molecules and controlling the intersystem crossing. Detailed experiments and theoretical calculation were performed to disclose the intrinsic relationship between the molecular structural characteristics (including molecular packing mode, torsion angle and the number of nitrophenyl groups, etc.) and the efficient/inefficient intersystem crossing. The data are detailed and the conclusions are reliable. Based on the solid research results, applications of this kind of material on optical information storage and haptic sensor are also exhibited. This is an elaborate piece of work. It puts forward a new train of thought on constructing ON/OFF mechanofluorochromic system, which is general, universal and applicable to other molecules. Thus, this work is recommended to be published on Nature Communications after some minor revisions:

1. The authors summed up traditional strategies on ON/OFF MRL materials as: “Few limited representatives of ON-OFF MRL materials utilize the photodimerization³⁰, photoinduced electron transfer^{31,32} and intramolecular charge transfer^{33,34} processes to control the solid-state luminescence. ...” Here, an important mechanism of strong π - π interactions modulation “RSC Adv., 2016, 6, 90305-90309; Chem. Asian J. 2016, 11, 3205– 3212.” should not be neglected.
2. In the experiments of morphology-dependent singlet oxygen detection (Supplementary Figure 2). The concentration of amorphous and nanocrystalline aggregates in the final solutions should be kept consistent.
3. In the application demonstration part, handed heat gun was used to convert amorphous solid to crystalline one, how to ensure the adhesion of powders with the substrates instead of being blown away?

Some typos should be corrected, such as “gringding”.

Reviewer #2 (Remarks to the Author):

The authors has developed a good strategy to realized high-contrast and highly sensitive ON-OFF MRL materials by manipulating ISC process in AIE compounds, via introducing nitro groups to promote the ISC decay and compete with AIE characteristic. Good applications have

been found in optical recording and haptic sensing. The design strategy is creative and rational. The experimental data, explanation and results are substantial. I recommend its publication in Nature Communications after some revisions.

1) In the introduction Section, when the authors mentioned the “On the other hand, the morphology of organic molecules with twisted conformations can be easily modulated by mechanical stimulus”, some vital references should be provided.

2) In the introduction Section, “AIEgens often have such twisted propeller-like conformation such as triphenylamine (TPA) and tetraphenylethylene (TPE) (Figure 1c)”. Actually, AIEgens involves many other twisted molecular conformation, such as twisted intramolecular charge transfer compounds (The Journal of Physical Chemistry C, 2015, 119, 2133–2141), . cis–trans isomerized configuration (J. Am. Chem. Soc. 2005, 127, 14152–14153) and herringbone conformation (J. Phys. Chem. B 2005, 109, 13472–13482). These various types should also be included.

3) Page 5, “They exhibit bright green or yellow fluorescence as well as in thin-films and in aggregates, suggesting the ISC is blocked due to morphology change. (Supplementary Figure 1, Table 1-2)” Table 1-2 are not provided in the main text.

Responses to Reviewers' Comments:

Responses to the Comments and Suggestions of Reviewer 1

Dear Reviewer 1:

The reviewer commented that “This is an elaborate piece of work. It puts forward a new train of thought on constructing ON/OFF mechanofluorochromic system, which is general, universal and applicable to other molecules. Thus, this work is recommended to be published on Nature Communications after some minor revisions”. He/she also pointed out several minor issues of the manuscript.

We thank the reviewer for his/her recognition of this work and the nice advices he/she made, and we revised the manuscript accordingly. Below are our point-to-point responses to the reviewer’s comments.

Comments 1: *The authors summed up traditional strategies on ON/OFF MRL materials as: “Few limited representatives of ON-OFF MRL materials utilize the photodimerization³⁰, photoinduced electron transfer^{31, 32} and intramolecular charge transfer^{33, 34} processes to control the solid-state luminescence. ...” Here, an important mechanism of strong π - π interactions modulation “RSC Adv., 2016, 6, 90305-90309; Chem. Asian J. 2016, 11, 3205– 3212.” should not be neglected.*

Our Reply: Thanks for your careful revision. We follow the reviewer’s consideration, and revised our manuscript as follows: *“intramolecular charge transfer^{33,34} and strong π - π interactions modulation^{35,36} to control the solid-state luminescence.”* The two important references have been added to our revised manuscript.

Comments 2: *In the experiments of morphology-dependent singlet oxygen detection (Supplementary Figure 2). The concentration of amorphous and nanocrystalline aggregates in the final solutions should be kept consistent.*

Our Reply: Thanks for your kind suggestion. The solution of amorphous aggregates has been changed to the same concentration as the solution of nanocrystalline aggregates. Then, we did the similar experiment of singlet oxygen detection and the same results were got which are shown in the revised Supplementary Information part (Supplementary Fig. 2).

Comments 3: *In the application demonstration part, handed heat gun was used to convert amorphous solid to crystalline one, how to ensure the adhesion of powders with the substrates instead of being blown away? Some typos should be corrected, such as “gringding”.*

Our Reply: Thanks for your careful revision. After thermal treatment by heat gun, the uniform crystalline film can adhere tightly on the surface of different substrates and the crystalline film cannot be blown away easily. There are may be two reasons: First, TPE-4N exhibits good film-forming and easy crystallization abilities (Figure 4), therefore the uniform spherulitic crystalline film can form quickly on the surface of substrates instead of being blown away; Second, the amorphous and crystalline film can adhere tightly on the surface because of the strong Van der Waals force between the molecule's nitro groups and substrates, On other hand, the switching experiments can be repeated for many times without any change and the applications are rewritable or reusable also exclude the possibility of blowing away the powders.

Some typos have been corrected in the revised manuscript.

Responses to the Comments and Suggestions of Reviewer 2

Dear Reviewer 2:

The reviewer commented that “The authors has developed a good strategy to realized high-contrast and highly sensitive ON-OFF MRL materials by manipulating ISC process in AIE compounds, via introducing nitro groups to promote the ISC decay and compete with AIE characteristic. Good applications have been found in optical recording and haptic sensing. The design strategy is creative and rational. The experimental data, explanation and results are substantial. I recommend its publication in Nature Communications after some revisions.” He/she also pointed out several minor issues of the manuscript.

We thank the reviewer for his/her recognition of this work and the nice advices he/she made, and we revised the manuscript accordingly. Below are our point-to-point responses to the reviewer’s comments.

Comments 1: *In the introduction Section, when the authors mentioned the “On the other hand, the morphology of organic molecules with twisted conformations can be easily modulated by mechanical stimulus”, some vital references should be provided.*

Our Reply: Thanks for your kind suggestion. Four important references have been added to the revised manuscript.

Comments 2: *In the introduction Section, “AIEgens often have such twisted propeller-like conformation such as triphenylamine (TPA) and tetraphenyethylene (TPE) (Figure 1c)”. Actually, AIEgens involves many other twisted molecular conformation, such as twisted intramolecular charge transfer compounds (The Journal of Physical Chemistry C, 2015, 119, 2133–2141), . cis–trans isomerized configuration (J. Am. Chem. Soc. 2005, 127, 14152–14153) and herringbone conformation (J. Phys. Chem. B 2005, 109, 13472–13482). These various types should also be included.*

Our Reply: Thanks for your kind suggestion. We follow the reviewer’s consideration, and revised our manuscript as follows: *“AIEgens often have twisted molecular conformation, such as*

twisted intramolecular charge transfer conformation, cis-trans isomerized configuration and herringbone conformation etc. Among them, triphenylamine (TPA) and tetraphenyethylene (TPE) are widely used as AIE cores with propeller-like conformation (Fig. 1c).” And three important references have also been added to the revised manuscript.

Comments 3: *Page 5, “They exhibit bright green or yellow fluorescence as well as in thin-films and in aggregates, suggesting the ISC is blocked due to morphology change. (Supplementary Figure 1, Table 1-2)” Table 1-2 are not provided in the main text.*

Our Reply: Thanks for your careful revision. We are sorry for the misleading description and we have changed it into “Supplementary Table 1-2”.